# Comparative Analysis of Sectional Anatomy, Computed Tomography and Magnetic Resonance of the Cadaveric Six-Banded Armadillo (*Euphractus sexcintus*) Head

**DOI:** 10.3390/vetsci12050433

**Published:** 2025-05-01

**Authors:** José Raduan Jaber, Daniel Morales-Bordon, Manuel Morales, Pablo Paz-Oliva, Mario Encinoso, Inmaculada Morales, Natalia Roldan-Medina, Gregorio Ramírez Zarzosa, Alejandro Morales-Espino, Alvaro Ros, Magnolia Maria Conde-Felipe

**Affiliations:** 1Departamento de Morfología, Facultad de Veterinaria, Universidad de Las Palmas de Gran Canaria, Trasmontaña, Arucas, 35413 Las Palmas, Spain; pablo.paz101@alu.ulpgc.es (P.P.-O.); natalia.roldan101@alu.ulpgc.es (N.R.-M.); alejandro.morales108@alu.ulpgc.es (A.M.-E.); alvaro.ros101@alu.ulpgc.es (A.R.); 2VETFUN, Educational Innovation Group, University of Las Palmas de Gran Canaria, Trasmontaña, Arucas, 35413 Las Palmas, Spain; mmconde@uco.es; 3Departamento de Patología Animal, Producción Animal, Bromatología y Tecnología de Los Alimentos, Facultad de Veterinaria, Universidad de Las Palmas de Gran Canaria, Trasmontaña, Arucas, 35413 Las Palmas, Spain; daniel.morales@ulpgc.es (D.M.-B.); mencinoso@gmail.com (M.E.); inmaculada.morales@ulpgc.es (I.M.); 4Departamento de Anatomía y Anatomía Patológica Comparadas, Facultad de Veterinaria, Universidad de Murcia, 30100 Murcia, Spain; grzar@um.es; 5Departamento de Sanidad Animal, Facultad de Veterinaria, Universidad de Córdoba, 14014 Córdoba, Spain

**Keywords:** computed tomography, magnetic resonance imaging, diagnostic imaging techniques, cross-sectional anatomy, head, central nervous system, six-banded armadillo

## Abstract

The six-banded armadillo (*Euphractus sexcintus*), like many other wildlife species, is classified as least concern by the International Union for Conservation of Nature (IUCN), primarily due to its broad distribution and large population size. However, in recent years, its population has declined in certain areas due to human activities. The limited literature available on the anatomy of this species prompted us to investigate the head of the six-banded armadillo by using modern imaging techniques, such as computed tomography and magnetic resonance imaging, combined with anatomical cross-section, to acquire valuable information about the structures comprising its head.

## 1. Introduction

*Euphractus sexcinctus*, commonly known as the six-banded armadillo, is a member of the Chlamyphoridae family [1]. It is one of the largest armadillo species, reaching approximately 40 centimeters of body length. This terrestrial and solitary mammal inhabits open areas, such as plains or savannahs, as well as forests and jungles. This animal can be found predominantly in the eastern regions of South America, particularly in countries like Argentina, Brazil, Uruguay, Paraguay, Bolivia, and certain areas of Suriname [2,3].

The six-banded armadillo presents dermal plates that cover most of its dorsum, including the head and tail. The carapace comprises six to eight flexible bands and shows a light coloration that varies from yellow to reddish brown [2]. The triangular, pointed-shaped, and flat head of this animal features a distinctive mosaic-like structure, and it has up to twenty-five teeth, which are practical for its omnivorous diet that includes invertebrates, carrion, and plants [2]. Both its *manus* and *pes* have five toes, which are used to dig inverted U-shaped burrows and small tunnels [2]. The gestation period for this species lasts about two months, during which they typically have one to three offspring. These young armadillos quadruple their weight within the first month and reach adulthood by nine months [1,2,3].

The six-banded armadillo is considered a least concern by the IUCN. However, it faces several risks, primarily due to extensive hunting for local use, including as a protein source and for handicraft and medicinal purposes [3]. With the increasing presence of exotic companion animals and the need to preserve wildlife, biologists, veterinarians, and conservationists must understand the complexities of these animals’ anatomy, physiology, and lifestyle to be able to diagnose faster and more accurately [4].

The significant anatomical differences among terrestrial mammals and the increasing interest in exotic animals have created challenges for veterinary clinicians in interpreting diagnostic imaging studies. Traditional imaging methods like radiology and ultrasound [4], along with advanced imaging techniques, including computed tomography (CT) and magnetic resonance imaging (MRI) [5,6,7,8,9,10,11], have greatly enhanced anatomical knowledge and pathology detection in veterinary medicine. Compared to traditional methods, modern imaging techniques provide superior resolution of anatomical structures, more precise definitions of lesion extent and characteristics, faster image acquisition, and elimination of superimposition. These advantages have revolutionized research, veterinary practice, and education [6,7,8,9,10,11,12,13,14,15,16,17,18,19,20,21,22,23,24,25,26,27,28,29,30,31,32,33,34,35,36,37,38,39,40,41].

Various studies have investigated the anatomy, physiology, and pathology of these species, particularly focusing on the nine-banded armadillo [12,13,14,15,16,17,18,19,20]. Research has examined various aspects, including the anatomy and functional morphology of xenarthrous vertebral processes, spinal development, skeletal and cranial structures, the auditory system, the nasal cavity and paranasal sinuses, the morphology of laryngeal cartilages, salivary glands, masticatory adaptations, and the visual cortex [12,13,14,15,16,17,18,19,20,21,22,23,24,25,27,33,34,35,36,37,38,39,42,43]. Additionally, other studies have addressed pathological conditions, such as fractures [22] and osteoderm lesions [23]. To date, a few papers have investigated different formations of the armadillo’s head [16,18,19,20,21,24,25,27,33,34,35,36,37,38,39,43]. However, no previous morphological studies using advanced imaging techniques have been developed to identify the central nervous system (CNS) and peripheral organs of the six-banded armadillo. Therefore, this research aimed to identify and describe the most obvious structures that comprise the six-banded armadillo’s head using cross-sectional anatomy, CT, and MRI. These imaging techniques, combined with detailed anatomical sections, could significantly enhance our understanding of the biology of the six-banded armadillo, providing insights into the evolutionary adaptation of armadillos and their phylogenetic relationships within the superorder Xenarthra. Moreover, this anatomical knowledge could help veterinarians in facilitating the detection and management of common conditions observed in these and related species, such as dental abscesses, malocclusions, diseases of the nasal cavity and paranasal sinuses, cranial fractures, and neurological disorders. Such conditions often require precise anatomical reference for accurate diagnosis and effective treatment planning.

## 2. Materials and Methods

### 2.1. Animals

Three adult six-banded armadillo carcasses—two males (designated as Armadillo 1 and Armadillo 2) and one female (Armadillo 3) from the species *Euphractus sexcintus*—were obtained from the Rancho Texas Lanzarote Park zoological facility (Canary Islands, Spain). All the armadillos evaluated were adults, which was confirmed based on the absence of visible cranial sutures in the imaging techniques. In addition, no age-related changes such as ventricular enlargement or cortical thinning were observed, suggesting that the animals were not of advanced age.

The animals died from causes unrelated to the central nervous system, and no pathological findings were evident upon clinical examination of the head. The absence of visible lesions and the good preservation of the carcasses offered an excellent opportunity for anatomical and imaging-based investigation. Although the precise postmortem interval was not recorded in hours, the specimens were promptly stored and handled under appropriate conditions to preserve tissue integrity and ensure optimal image quality. One week after death, carcasses were kept out of the freezer for 24 h prior to imaging procedures, allowing adequate thermal equilibration and minimizing temperature-related imaging artifacts.

### 2.2. CT Technique

To acquire detailed CT images, we employed a 16-slice helical CT scanner (Toshiba Astelion, Canon Medical System, Tokyo, Japan) at the Veterinary Hospital of the University of Las Palmas de Gran Canaria. The armadillos were carefully positioned in symmetrical ventral recumbency symmetrically on the CT table to ensure optimal imaging. We captured sequential CT images with a 1 mm slice thickness following a standard protocol (100 kVp, 80 mA, 512 × 512 acquisition matrix, 1809 × 858 field of view, a spiral pitch factor of 0.94, and a gantry rotation time of 1.5 s). To enhance visualization of the head structures, three CT window settings with varying widths and levels were applied: a bone window (WW = 1500; WL = 300), a lung window (WW = 1400; WL = −500), and a soft tissue window (WW = 350; WL = 40), focusing on the bone window images for clarity. Furthermore, dorsal and sagittal multiplanar reconstructions (MPR) were generated to facilitate detailed evaluation of the complex head structures. All CT images were subsequently imported into an advanced image platform (OsiriX MD, Apple, Cupertino, CA, USA) for post-processing, ensuring a comprehensive evaluation of these unique specimens.

### 2.3. MRI Technique

Magnetic resonance imaging (MRI) was performed on three six-banded armadillo specimens immediately following the CT examination, using a 1.5-Tesla magnet (Toshiba, Vantage Elan, Tokyo, Japan). The animals were maintained in ventral recumbency to ensure consistent anatomical orientation. A standard MRI protocol was followed to acquire spin-echo (SE) T1-weighted and T2-weighted images in transverse, sagittal, and dorsal planes. For SE T1-weighted transverse images, the parameters were an echo time (TE) of 10 ms, a repetition time (TR) of 800 ms, a matrix size of 536 × 384, and a slice thickness of 4.5 mm with 4 mm spacing between slices. The SE T2-weighted transverse images were obtained with a TE of 120 ms, TR of 10,541 ms, matrix size of 624 × 448, and a slice thickness of 3 mm with 3 mm spacing. The SE T2-weighted sagittal images had a TE of 120 ms, TR of 7529 ms, matrix size of 512 × 804, and a slice thickness of 2.8 mm with 2 mm interslice spacing. For SE T2-weighted dorsal images, the parameters were a TE of 120 ms, a TR of 8282 ms, a matrix size of 468 × 512, and a slice thickness of 3.4 mm with 3 mm interslice spacing. We used a medical imaging viewer (OsiriX MD, Geneva, Switzerland) to analyze the acquired images.

### 2.4. Anatomical Sections

After the scanning procedure, all three six-banded armadillos were placed in dorsal recumbency within expanded polystyrene containers and rapidly frozen at −80 °C for 72 h. Subsequently, serial anatomical sections approximately 0.5 cm thick were obtained using an electric band. Each slice was promptly rinsed with water, labeled numerically, and photographed from both sides.

### 2.5. Anatomic Evaluation

The sections that best corresponded with the CT and MR images were selected to assist in identifying key structures of the six-banded armadillo’s head. Additionally, consultation with textbooks and relevant references on the anatomy of armadillos and other mammals was required [21,24,25,26,27,28,29,30,31,32,33,34,35,36,37,38,39,40,41,42,43,44,45,46,47,48,49,50,51]. We named most of these structures in compliance with the International Committee on Veterinary Gross Anatomical Nomenclature. English terms were used when referring to broader concepts, functional aspects, or when Latin equivalents are not commonly used in the literature or clinical context. This mixed approach aims to balance anatomical precision with readability and is consistent with the practice adopted in several recent anatomical and veterinary studies [24,25,26,27,28,29,30,31,32].

## 3. Results

Eight representative transverse sections of the armadillo head were selected (Figure 1). Figure 2, Figure 3, Figure 4, Figure 5, Figure 6, Figure 7, Figure 8 and Figure 9 comprise four images: (A) anatomic cross-section, (B) CT bone window, (C) T1-weighted (T1W), and (D) T2-weighted (T2W) MR images. These images are shown in a rostrocaudal progression from the olfactory bulb (Figure 2) to the brainstem levels (Figure 9). In addition, Figure 10 contains three images: a median CT bone window image (B), a median T1W MR image (C), and a paramedian T2W MR image (D). Finally, Figure 11 comprises a bone-window CT dorsal (B) and a dorsal MRI in T2W (D) images at the level of the tympanic cavity.

### 3.1. Anatomical Study

Pivotal formations of the head were identified by anatomical cross-sections. Hence, we observe the cerebrum divided into left and right telencephalic hemispheres by a deep longitudinal fissure. Each hemisphere was divided into a rostral frontal lobe, a caudal occipital lobe, a lateral temporal lobe, and a parietal lobe between frontal and occipital lobes (Figure 2A, Figure 3A, Figure 4A, Figure 5A, Figure 6A, Figure 7A and Figure 8A). Most cranial sections showed an excellent depiction of the greatly developed olfactory bulbs found rostrally in the cerebral hemispheres (Figure 2A). Here, we also distinguished enlarged olfactory recesses bounded medially by the perpendicular plate of the ethmoidal bone (Figure 2A and Figure 3A). The following sections helped distinguish the connexion of both telencephalic hemispheres by fibers of the white matter forming the corpus callosum (Figure 5A and Figure 6A). Moreover, these sections allowed the identification of a cavity in each of the hemispheres forming the lateral ventricles, which presented the caudate nuclei bulging into them (Figure 5A, Figure 6A, Figure 7A and Figure 8A).

Additionally, several components of the diencephalon could be depicted. Therefore, we observed that most of the diencephalon was occupied by the thalamus, mainly constituted by gray matter. The thalamus of both sides communicates at the midline, creating the adhesio interthalamica that obliterates the central space of the third ventricle. Other structures included the hypophysis and the lateral and medial geniculate bodies that receive visual and auditory stimuli (Figure 5A, Figure 7A, and Figure 8A). These formations were limited laterally by white matter fibers called the capsula interna. The ventral part of this capsule and the piriform lobes were essential remarks to identify the amygdala and the globus pallidus (Figure 6A). On the most caudal sections, we identified the mesencephalon, showing the aqueductus mesencephali, the pedunculus cerebri, and the colliculus rostralis (Figure 8A). From the metencephalon, we could locate the cerebellar hemispheres, the vermis, the nodulus covering part of the fourth ventricle, and prominent ventral structures, including the pons and the transverse pontine fibers forming the pedunculus cerebellaris medius (Figure 9A). In addition, these sections provide relevant insights into the myelencephalon, containing the fourth ventricle and the presence of bilateral pyramis on its ventral surface (Figure 9A).

Regarding the bony formations, different bones comprising the skull were identified, including the frontal, temporal (with its squamous, petrous, and tympanic components), the zygomatic arch, the sphenoid, the maxilla, the mandible, and the occipital bones (Figure 2A, Figure 3A, Figure 4A, Figure 5A, Figure 6A, Figure 7A, Figure 8A and Figure 9A). Additionally, these images facilitated the visualization of structures of the cavum oris, including the tongue, the palatine plexus, and the soft palate (Figure 2A, Figure 3A, Figure 4A, Figure 5A, Figure 6A, Figure 7A and Figure 8A), and other important components of the auditory system, such as the external ear with the acoustic meatus and the cartilago auriculae (Figure 9A). Moreover, the tympanic cavity and bulla were also depicted in these sections (Figure 7A, Figure 8A and Figure 9A). Furthermore, we distinguished specific masticatory muscles, including the musculus temporalis, the musculi pterygoideus medialis and lateralis, and the musculus masseter (Figure 4A, Figure 5A, Figure 6A, Figure 7A and Figure 8A).

### 3.2. Computed Tomography Study

Concerning the nervous system, the CT images distinguished relevant formations by their relative position. Therefore, we identified the olfactory bulbs as symmetrical bilateral prominences with intermediate attenuation (Figure 2B, Figure 10B, and Figure 11B). However, we could not identify the olfactory recesses with this imaging technique due to its low resolution for soft tissues. Despite this limitation, we successfully identified the fissura longitudinalis cerebri separating the two telencephalon hemisphaerium (Figure 2B and Figure 3B).

Additionally, we defined other nervous structures with similar attenuation, including the lobi piriformi, which were recognized thanks to their globose shape and ventrolateral position (Figure 6B). Moreover, these sections depicted significant parts of the diencephalon, such as the hypophysis, resting in the pituitary fossa (Figure 8B). More caudally, the dorsal, transverse, and sagittal CT images identified relevant formations of the metencephalon, including the median region termed the vermis and the adjacent bilateral cerebellum hemisphaerum (Figure 9B, Figure 10B and Figure 11B). Moreover, these caudal sections helped in depicting the myelencephalon (medulla oblongata) (Figure 9B, Figure 10B and Figure 11B).

The CT images revealed pivotal skull bones showing high CT attenuation. Therefore, we distinguished the frontal, zygomatic process, vomer, occipital, basioccipital, sphenoid, temporal, and maxillary bones (Figure 2B, Figure 3B, Figure 4B, Figure 5B, Figure 6B, Figure 7B, Figure 8B, Figure 9B and Figure 10B). Moreover, we identified different parts of the mandible, including the ramus and the processus condylaris, the articulatio temporomandibularis, and some components of the larynx and the apparatus hyoideus (Figure 2B, Figure 3B, Figure 4B, Figure 5B, Figure 6B, Figure 7B, Figure 8B, Figure 9B, Figure 10B and Figure 11B). It is important to emphasize that the most notable bony structure of this animal, due to its species exclusivity, corresponds to the osteoderms, located across the entire dorsal portion of the head as a row of ossified geometric plates (Figure 2B, Figure 3B, Figure 4B, Figure 5B, Figure 6B, Figure 7B, Figure 8B, Figure 9B, Figure 10B and Figure 11B).

These CT images were also helpful in depicting essential elements of the auditory system, such as the external ear and the bulla tympanica, recognizable on both sides of the midline of the skull, laterally followed by the meatus acustici externi (Figure 8B, Figure 9B, and Figure 11B). Moreover, other anatomical structures with intraluminal content, such as the nasopharynx and the cavum oris, were displayed with this technique, showing a vacuum effect (Figure 2B, Figure 3B, Figure 4B, Figure 5B, Figure 6B, Figure 7B, Figure 8B, Figure 9B and Figure 10B).

Finally, several muscles were recognized with intermediate attenuation, such as the masseter and the temporal muscles (Figure 2B, Figure 3B, Figure 4B, Figure 5B, Figure 6B, Figure 7B and Figure 8B).

### 3.3. Magnetic Resonance Imaging (MRI)

No consistent anatomical differences were distinguished in the six-banded armadillos studied, and the anatomical sections satisfactorily matched with the structures shown in the MRI images. Furthermore, compared to CT, the MRI technique portrayed a better view of the CNS structures. Therefore, the use of this technique allowed the identification of the olfactory bulbs (Figure 2C,D, Figure 10C,D, and Figure 11D) as triangular, rostral structures with a moderate to hyperintense uniform signal intensity, divided by the *fissura longitudinalis cerebri* (Figure 2C,D, Figure 4C,D, Figure 5D, Figure 6D, Figure 7C,D, and Figure 8D). Their identification was crucial to the olfactory recess observation (Figure 2D, Figure 3D, Figure 4D, Figure 10D and Figure 11D), which appears as hyperintense in the T2W images, although they were not visible in the T1W images. These more rostral images also facilitated the identification of the *sinus sagittalis dorsalis* (Figure 2C,D, Figure 3C,D, Figure 4C,D, Figure 5C,D, Figure 6C,D, Figure 7C,D and Figure 8C,D), represented as moderately intense and hyperintense in T1W and T2W images, respectively. Furthermore, the *telencephalon hemisphaerium* was observed in both T1W and T2W, with moderate and regular intensity signals (Figure 3C,D, Figure 4C,D, Figure 5C,D, Figure 6C,D, Figure 7C,D, Figure 8C,D, Figure 9C,D, Figure 10C,D, and Figure 11D).

Moreover, the transverse, median, and dorsal images depicted relevant components of the ventricular system, including the lateral ventricles, the dorsal and ventral parts of the third ventricle, the *aqueductus mesencephali*, and the fourth ventricle, displaying moderate intense and hyperintense signals in T1W and T2W images, respectively (Figure 5C,D, Figure 6C,D, Figure 7C,D, Figure 8C,D, Figure 9D, Figure 10C,D, and Figure 11D). The lateral ventricles facilitated the observation of the *nucleus caudatus* nucleus, shown with a moderate intensity and an oval shape (Figure 5C,D, Figure 10D, and Figure 11D), and the *hippocampus*, a large bean-shaped structure with a moderate intensity signal in both T1W and T2W images (Figure 6C,D, Figure 7C,D, Figure 8C,D, Figure 10D, and Figure 11D). In addition, the two parts of the third ventricle enclosed the *adhesio interthalamica*, limited laterally by the right and left sides of the *thalamus* (Figure 6D, Figure 7D, and Figure 10C,D). Pivotal white matter identifiable structures were the *corpus callosum*, the *fornix*, and the *capsula interna*, with a lower intensity than other structures in the CNS, but only visible in T2W (Figure 5D, Figure 6D, Figure 7D, Figure 8D, Figure 10D, and Figure 11D). A slight hypointense layer of white matter was essential to demarcate the *amygdala*, showing a moderate intensity signal (Figure 6C,D).

On the more caudal transverse and sagittal MR images, we distinguished the *colliculi* (with exceptional development of the *colliculus caudalis*) and the *pedunculus cerebri* with adequate resolution (Figure 8D and Figure 10C,D). Close to these peduncles, we could visualize the genicular bodies, showing intermediate signals (Figure 7C,D, Figure 8D, and Figure 10D). Other relevant formations of the nervous system, including the *vermis* and the *cerebellum hemisphaerium*, the *pedunculus cerebellaris medius*, and the *medulla oblongata*, were displayed in the T1 and T2W images (Figure 9C,D, Figure 10C,D, and Figure 11D).

MRI images were also pivotal to appreciate diverse eyeball components, such as the *humor vitreus* and *lens*, and the extraocular muscles (Figure 2C,D and Figure 3C,D), displaying moderate or high intensity signals. Additionally, we accurately pinpointed different muscles responsible for the chewing movement of the mandible and the *articulatio temporomandibularis*, including the lateral and medial pterygoid muscles and the masseter and temporal muscles, which presented moderate intensity signals in T1 and T2W transverse images (Figure 4C,D, Figure 5C,D, Figure 6C,D, Figure 7C,D and Figure 8C,D).

## 4. Discussion

The use of advanced imaging techniques in exotic animal and wildlife medicine remains limited due to their high cost, availability, and logistical challenges in obtaining images from certain species. While CT and MRI studies of the CNS have been conducted in both exotic and domestic animals, including dogs, horses, iguanas, porcupines, guinea pigs, rabbits, and the nine-banded armadillo [10,25,26,27,28,29,30,31], there is a lack of detailed anatomical descriptions of these images for the normal six-banded armadillo. Therefore, we performed a comprehensive anatomical analysis of the six-banded armadillo head through the combination of anatomical cross-section and CT and MRI images.

In this investigation, due to the limited number of specimens, we did not perform sagittal and dorsal anatomical sections because we were focused on obtaining those that better matched the transverse CT and MR images. Nevertheless, the transverse anatomical sections proved highly effective for identifying the main formations of the six-banded armadillo head. Hence, these images facilitated the identification of pivotal structures of its *encephalon*, with special attention to the exceptional development of the rhinencephalic structures and formations belonging to the oral cavity and the auditory system. However, its eyes were not discernible because of its small size and position, a common anatomical characteristic described in higher or lesser degrees by other xenarthrans [51]. Despite this limitation, they were clearly defined in the CT and MR images.

Median and dorsal CT images provided a clear depiction of the triangular, pointed-shaped, and flat head of this animal, consistent with descriptions in other species of the superorder Xenarthra, such as anteaters [51]. In addition, since the air provided excellent contrast in transverse CT images, the *cavum oris*, the nasopharynx, and the tympanic portion of the temporal bone were displayed due to the different contrast [30]. Notably, the marked development of the tympanic bullae could be relevant for explaining its auditory capacities. Similar findings have also been reported in other species, including the guinea pig, the crested porcupine, and other phylogenetically related xenarthrans [10,30,51]. Moreover, the CT bone window was especially useful in identifying various bony structures of the head, including the osteoderms, the zygomatic arch, the frontal, the squamous and petrous parts of the temporal, the occipital bone, the maxilla, and the mandible. In contrast, it exhibited a low capacity for identifying soft tissue like muscles or nervous tissue due to the minor density differences in CT images. Identical findings were displayed in investigations performed on other exotic species, such as the rhinoceros iguana, the guinea pig, the six-banded armadillo, rabbits, and the crested porcupine [24,28,30,32].

Although individual skull bones were not clearly differentiated on the MR images, they could be distinguished by their characteristic hypointense signal, consistent with previous reports [32]. In contrast to CT, both T1W and T2W MR images displayed superior definition of the nervous system, aligning with other studies that highlight MRI as the preferred imaging technique for brain imaging in veterinary and comparative anatomy [4,5,7,10,11,28,30,32]. T1W MR images offered greater definition of bone contours and soft tissue details, while T2W MR images were invaluable for neuroanatomical assessment. Notably, T2W MR images were crucial to visualize the great development of the rhinencephalic formations, including the olfactory bulbs and recesses, as well as the intermediate and lateral olfactory tracts. These features have been related to a well-developed sense of smell by improving olfactory airflow and odorant retention within the olfactory region [45,46]. This is consistent with the ecological behavior of Euphractus sexcinctus, which relies heavily on olfaction for foraging and predator detection, as documented in prior studies [2,3,47]. While our assessment was descriptive, the relative size and organization of the olfactory bulbs appear proportionally larger compared to many generalist mammals and are in line with what is expected in species that depend heavily on their sense of smell. Similar adaptations have been described in the seven-banded armadillo [47] and to a lesser extent in carnivores [48], rodents [49], and ungulates [50]. Other pivotal nervous formations were also pinpointed, including the *hippocampus*, *nucleus caudatus*, *colliculus caudalis*, and *amygdala*, further highlighting the value of MRI for detailed brain analysis.

Subjective image analysis and objective measurements identified a large *amygdala*. Studies performed on *Dasypus hybridus* have also identified a relevant development of this formation [47], mainly referring to the cortical and medial nuclei of the amygdaloid complex. This finding could be associated with the notable development of the *rhinencephalon*, which has been demonstrated in other animals [47]. Interestingly enough, the poor eyesight these animals exhibit leads them to rely on their sense of smell [2,3]. This sense is quite helpful in detecting prey and predators, and the amygdala can produce a fast response against predators, such as rolling into a ball covering their vital areas with armor or burrow digging in 45 **s** [52].

It is also crucial to note that our study, which mainly focuses on the nervous system, is limited by including only three armadillo specimens. However, it is essential to highlight that other investigations on xenarthrans have also relied on a limited number of specimens to explore various anatomical aspects. These include studies on the petrosal anatomy, nasal cavity, and laryngeal cartilage morphology of the nine-banded armadillo; the orientation selectivity in the visual cortex of the nine-banded armadillo; the macroscopic anatomy of the central nervous system in both six- and nine-banded armadillos; and the identification of head structures in the giant anteater [16,18,20,25,27,47,51]. Moreover, most of the studies on these species were performed in countries where these animals can be found, which can facilitate the obtention of specimens. Despite this, our study could be an initial reference for clinicians, biologists, and researchers. Furthermore, to gain a more comprehensive understanding of these findings and to expand upon the information presented in studies like this, additional research involving more specimens and a deeper understanding of the species’ natural history is necessary.

Finally, CT and MRI proved to be complementary imaging modalities. CT offered a rapid and detailed evaluation of bones, sinus cavities, and other air-containing spaces, which are essential in forensic and pathological investigations [53]. CT also served as a useful reference to guide subsequent MRI acquisition and facilitate precise anatomical correlation. In contrast, MRI provided superior soft tissue contrast, especially for CNS evaluation. It allowed detailed visualization of brain structures with sufficient detail to differentiate gray and white matter, ventricular morphology, and even subtle anatomical landmarks. While CT offered structural context, MRI was essential for assessing neural components not visible on CT. Together, these techniques represent essential tools not only for anatomical and diagnostic purposes but also for planning surgical interventions and monitoring therapeutic outcomes in this species.

## 5. Conclusions

Using anatomical cross-sections combined with modern diagnostic techniques was pivotal in accurately identifying the main structures of the armadillo head. Hence, the CT study was quite helpful in displaying specific morphological characteristics of the armadillo’s head, mainly those related to the different components of the skull. Moreover, brain formations were not easily identified due to their homogeneous appearance on CT images. However, T1W and T2W MR displayed the encephalic structures with adequate resolution. The results presented here could be valuable for assessing various pathological conditions, including skull fractures, injuries produced during mechanical immobilization for clinical examination, and neurological conditions, such as convulsions and tumors [54]. Additionally, these techniques can enhance veterinary anatomy education by providing a clear view of structures without the interference of overlapping parts, making it easier to visualize the extent of different types of lesions.

## Figures and Tables

**Figure 1 vetsci-12-00433-f001:**
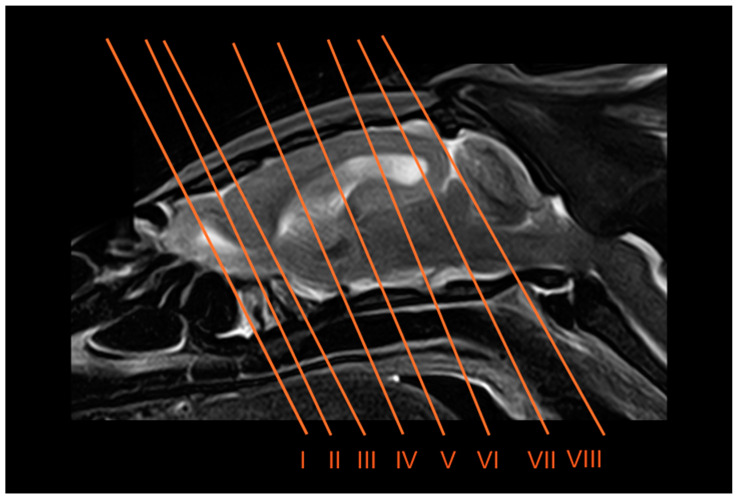
Paramedian T2W MR image of the six-banded armadillo showing the approximate levels of transverse slices. Each number represents the approximate location for each CT and MR transverse (I to VIII) images.

**Figure 2 vetsci-12-00433-f002:**
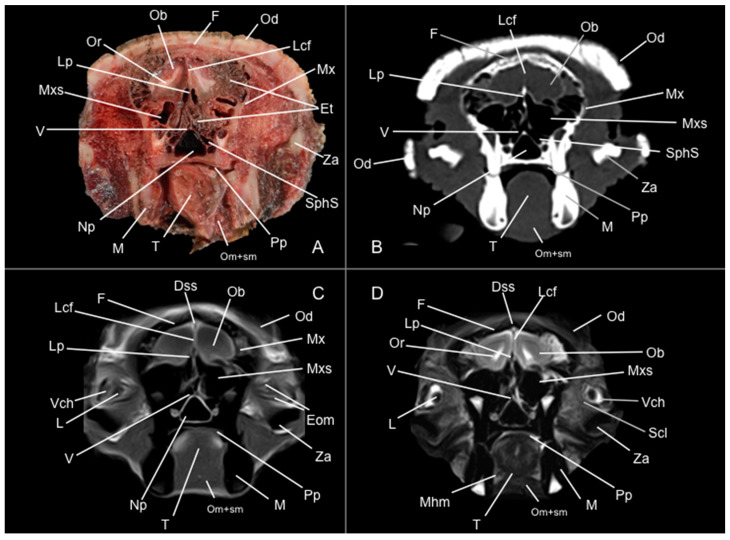
Anatomical section (**A**), CT bone window (**B**), T1W (**C**), and T2W (**D**) MR images of the head of a six-banded armadillo (armadillo 1) at the level of the olfactory bulb, corresponding to line I in Figure 1. Dss: sinus sagittalis dorsalis. Eom: extraocular muscles. Et: ethmoturbinalia. F: os frontale. L: lens. Lcf: fissura longitudinalis cerebri. Lp: lamina perpendicularis (os ethmoidale). M: mandible (body). Mhm: musculus mylohyoideus. Mx: maxilla. Mxs: sinus maxillaris. Np: nasopharynx. Ob: olfactory bulb. Od: osteoderm. Om+sm: musculi omohyoideus+ sternohyoideus. Or: olfactory recess. Pp: plexus palatinus. Scl: sclera. SphS: sinus sphenoidalis. T: lingua. V: vomer. Vch: camera vitrea bulbi. Za: arcus zygomaticus.

**Figure 3 vetsci-12-00433-f003:**
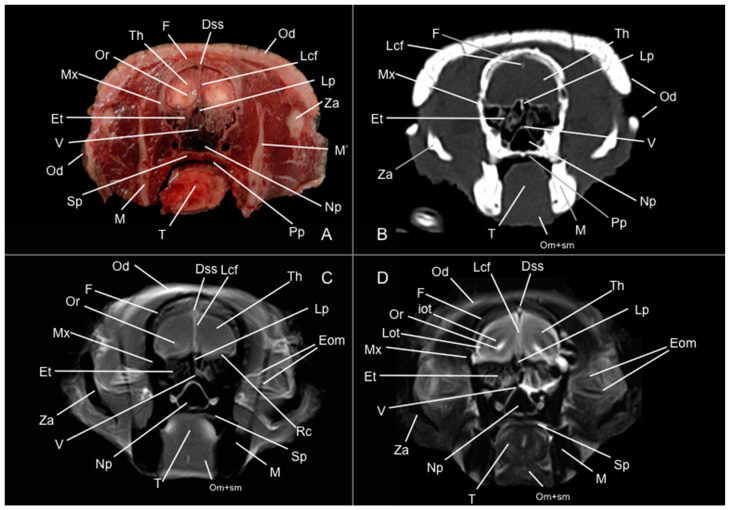
Anatomical section (**A**), CT bone window (**B**), T1W (**C**), and T2W (**D**) MR images of the head of a six-banded armadillo (armadillo 1) at the level of the olfactory recess, corresponding to line II in Figure 1. Dss: sinus sagittalis dorsalis. Eom: extraocular muscles. Et: ethmoturbinalia. F: os frontale. iot: tractus olfactorius intermedius. Lcf: fissura longitudinalis cerebri. Lot: tractus olfactorius lateralis. Lp: lamina perpendicularis (os ethmoidale). M: mandible (body). M’: mandible (ramus). Mx: maxilla. Np: nasopharynx. Od: osteoderm. Om+sm: musculi omohyoideus + sternohyoideus. Or: olfactory recess. Pp: plexus palatinus. Rc: commissura rostralis. Sp: soft palate. T: lingua. Th: telencephalon hemispherium (frontal lobe). V: vomer. Za: arcus zygomaticus.

**Figure 4 vetsci-12-00433-f004:**
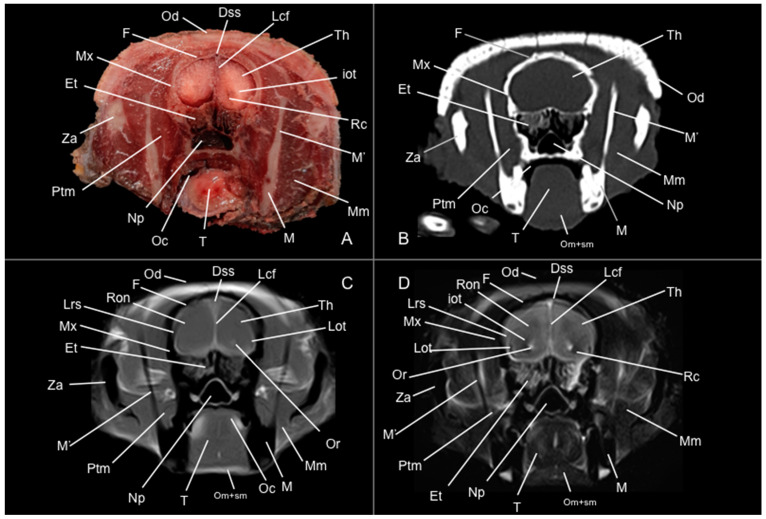
Anatomical section (**A**), CT bone window (**B**), T1W (**C**), and T2W (**D**) MR images of the head of a six-banded armadillo (armadillo 1) at the level of the rostral commissure, corresponding to line III in Figure 1. Dss: sinus sagittalis dorsalis. Et: ethmoturbinalia. F: os frontale. iot: tractus olfactorius intermedius. Lcf: fissura longitudinalis cerebri. Lot: tractus olfactorius lateralis. Lrs: sulcus rhinalis lateralis. M: mandible (body). M’: mandible (ramus). Mm: musculus masseter. Mx: maxilla. Oc: cavum oris. Od: osteoderm. Om+sm: musculi omohyoideus + sternohyoideus. Or: olfactory recess. Ptm: musculus pterygoideus medialis. Rc: commissura rostralis. Ron: rostral olfactory nucleus. T: lingua. Th: telencephalon hemispherium (frontal lobe). Za: arcus zygomaticus.

**Figure 5 vetsci-12-00433-f005:**
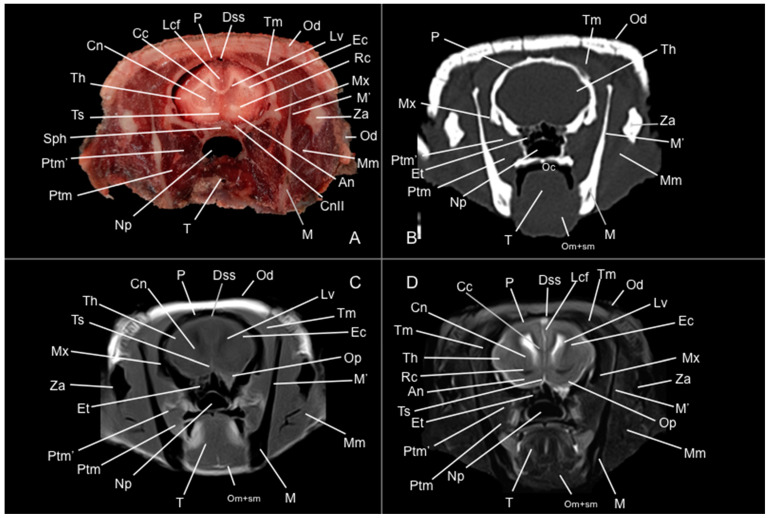
Anatomical section (**A**), CT bone window (**B**), T1W (**C**), and T2W (**D**) MR images of the head of a six-banded armadillo (armadillo 2) at the level of the body of the caudate nucleus, corresponding to line III in Figure 1. An: nucleus accumbens. Cc: corpus callosum. Cn: nucleus caudatus (body). CnII: cranial nerve II. Dss: sinus sagittalis dorsalis. Et: ethmoturbinalia. Ec: capsula externa. Lcf: fissura longitudinalis cerebri. Lv: lateral ventricle. M: mandible (body). M’: mandible (ramus). Mm: musculus masseter. Mx: maxilla. Np: nasopharynx. Oc: cavum oris. Od: osteoderm. Om+sm: musculi omohyoideus + sternohyoideus. Op: pedunculus olfactorius. P: os parietale. Ptm: musculus pterygoideus medialis. Ptm’: musculus pterygoideus lateralis. Rc: commissura rostralis. Sph: os sphenoidale. T: lingua. Th: telencephalon hemispherium (parietal lobe). Tm: musculus temporalis. Ts: septum telencephali. Za: arcus zygomaticus.

**Figure 6 vetsci-12-00433-f006:**
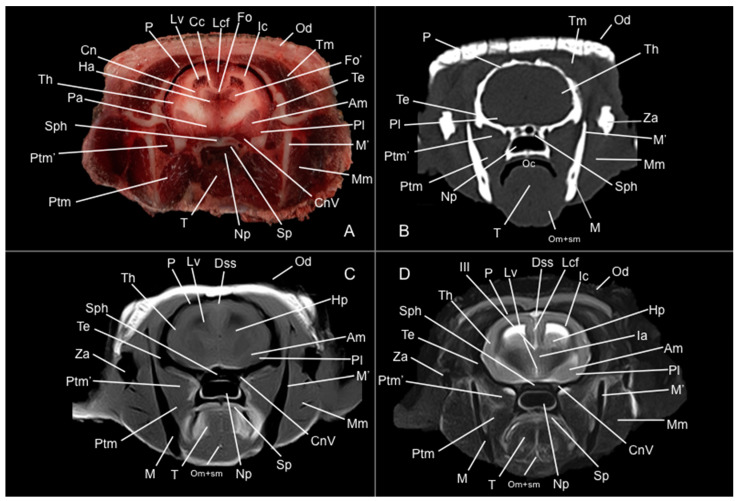
Anatomical section (**A**), CT bone window (**B**), T1W (**C**), and T2W (**D**) MR images of the head of a six-banded armadillo (armadillo 2) at the level of the amygdala corresponding to line V in Figure 1. Am: amygdala. Cc: corpus callosum. Cn: nucleus caudatus (tail). CnV: cranial nerve V. Dss: sinus sagittalis dorsalis. Fo: fornix. Fo’: pillars of fornix. Ha: habenula. Hp: hippocampus. Ic: capsula interna. III: third ventricle. Ia: adhesio interthalamica. Lcf: fissura longitudinalis cerebri. Lv: lateral ventricle. M: mandible (body). M’: mandible (ramus). Mm: musculus masseter. Np: nasopharynx. Oc: cavum oris. Od: osteoderm. Om+sm: musculi omohyoideus + sternohyoideus. P: os parietale. Pa: globus pallidus. Pl: lobus piriformis. Ptm: musculus pterygoideus medialis. Ptm’: musculus pterygoideus lateralis. Sp: soft palate. Sph: os sphenoidale. T: lingua. Te: os temporale. Th: telencephalon hemispherium (temporal lobe). Tm: musculus temporalis. Za: arcus zygomaticus.

**Figure 7 vetsci-12-00433-f007:**
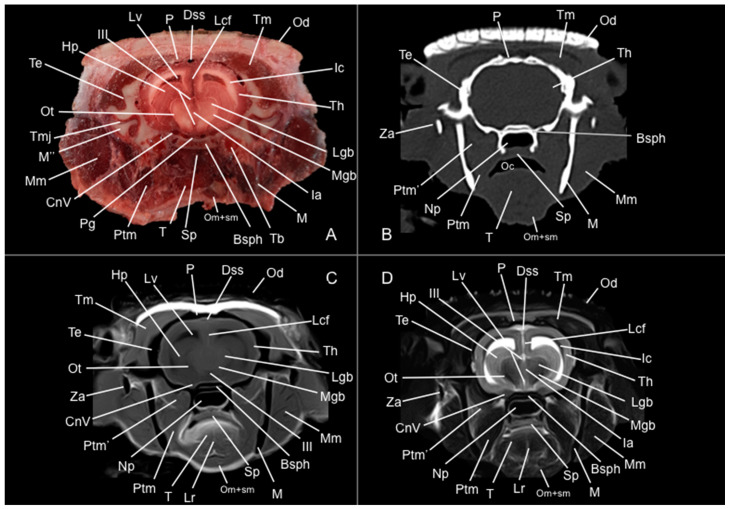
Anatomical section (**A**), CT bone window (**B**), T1W (**C**), and T2W (**D**) MR images of the head of a six-banded armadillo (armadillo 3) at the level of the caudal part of the diencephalon, corresponding to line VI in Figure 1. CnV: cranial nerve V. Dss: sinus sagittalis dorsalis. Hp: hippocampus. Ia: adhesio interthalamica. Ic: capsula interna. III: third ventricle. Lcf: fissura longitudinalis cerebri. Lgb: corpus geniculatum laterale. Lr: radix linguae. Lv: lateral ventricle. M’: mandible (ramus). M’’: processus condylaris (mandible). Mgb: corpus geniculatum mediale. Mm: musculus masseter. Np: nasopharynx. Oc: cavum oris. Od: osteoderm. Om+sm: musculi omohyoideus + sternohyoideus. Ot: tractus opticus. P: os parietale. Pg: hypophysis. Ptm: musculus pterygoideus medialis. Ptm’: musculus pterygoideus lateralis. Sp: soft palate. Bsph: os basisphenoidale. T: lingua. Tb: bulla tympanica. Te: os temporale. Th: telencephalon hemispherium (temporal lobe). Tm: musculus temporalis. TMj: articulatio temporomandibularis. Za: arcus zygomaticus.

**Figure 8 vetsci-12-00433-f008:**
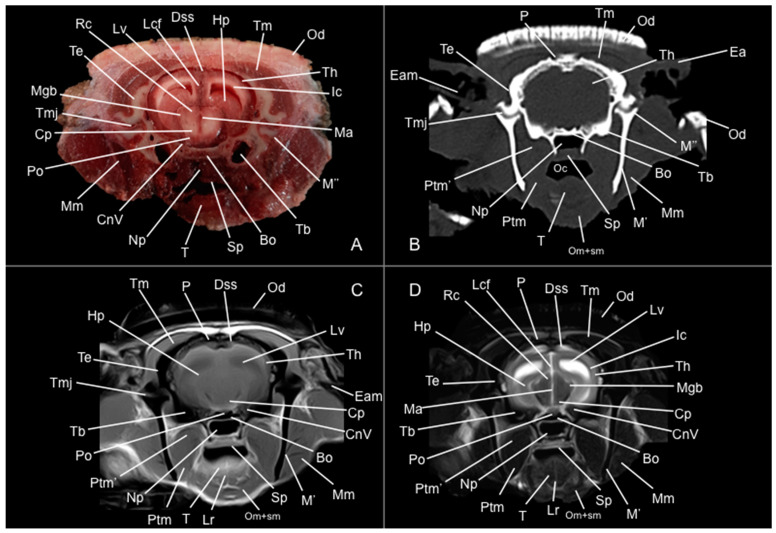
Anatomical section (**A**), CT bone window (**B**), T1W (**C**), and T2W (**D**) MR images of the head of a six-banded armadillo (armadillo 3) at the level of the mesencephalon corresponding to line VII in Figure 1. Bo: basioccipital. CnV: craneal nerve V. Cp: pedunculus cerebri. Dss: sinus sagittalis dorsalis. Ea: ear. Eam: meatus acustici externi. Hp: hippocampus. Ic: capsula interna. Lcf: fissura longitudinalis cerebri. Lr: radix linguae. Lv: lateral ventricle. M’: mandible (ramus). M’’: processus condylaris (mandible). Ma: aqueductus mesencephali. Mgb: corpus geniculatum mediale. Mm: musculus masseter. Np: nasopharynx. Oc: cavum oris. Od: osteoderm. Om+sm: musculi omohyoideus + sternohyoideus. P: os parietale. Po: pons. Ptm: musculus pterygoideus medialis. Ptm’: musculus pterygoideus lateralis. Rc: colliculus rostralis. Sp: soft palate. T: lingua. Tb: bulla tympanica. Te: os temporale. Th: telencephalon hemispherium (occipital lobe). Tm: musculus temporalis. Tmj: articulatio temporomandibularis.

**Figure 9 vetsci-12-00433-f009:**
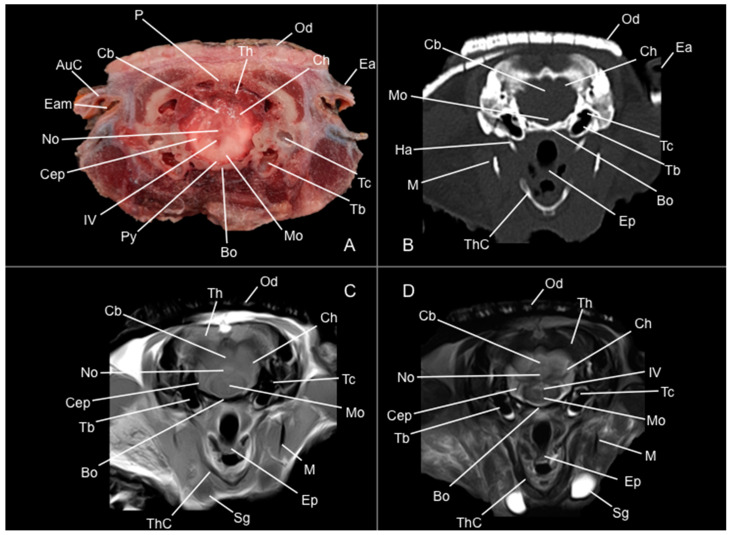
Anatomical section (**A**), CT bone window (**B**), T1W (**C**), and T2W (**D**) MR images of the head of a six-banded armadillo (armadillo 3) at the level of the fourth ventricle, corresponding to line VIII in Figure 1. AuC: cartilago auriculae. Bo: basioccipital. Cb: cerebellum (body). Cep: pedunculus cerebellaris (medius). Ch: cerebellum hemispherium. Ea: ear. Ep: epiglottis. Ha: apparatus hyoideus (os hyoideum). IV: fourth ventricle. M: mandible. Mo: medulla oblongata. No: nodulus. Od: osteoderm. Py: pyramis. Sg: glandulae salivariae. Tb: bulla tympanica. Tc: cavum tympani. Th: telencephalon hemispherium (occipital lobe). ThC: cartilago thyroidea.

**Figure 10 vetsci-12-00433-f010:**
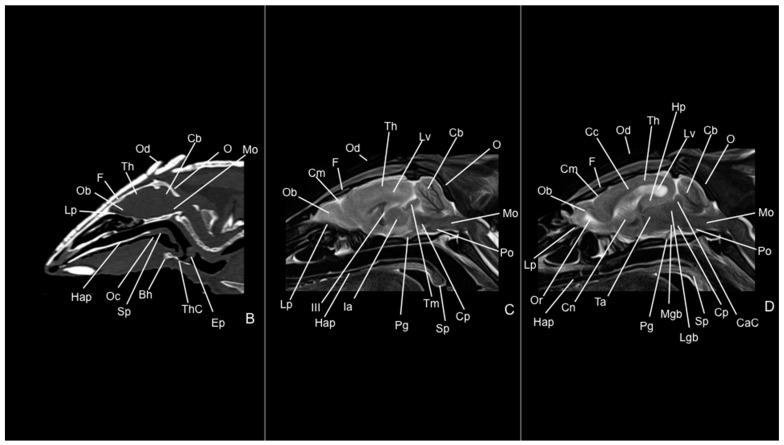
Median CT bone window (**B**), median T1W (**C**), and paramedian T2W (**D**) MR images of the head of a six-banded armadillo (armadillo 1). Bh: basihyoideum. CaC: colliculus caudalis. Cb: cerebellum (vermis). Cc: corpus callosum. Cm: cerebral meninges. Cn: nucleus caudatus. Cp: pedunculus cerebri. Ep: epiglottis. F: os frontale. Hap: hard palate. Hp: hippocampus. Ia: adhesio interthalamica. III: third ventricle. Lgb: corpus geniculatum laterale. Lp: lamina perpendicularis (os ethmoidale). Lv: lateral ventricle. Mgb: corpus geniculatum mediale. Mo: medulla oblongata. O: os occipitale. Ob: olfactory bulb. Oc: cavum oris. Od: osteoderm. Or: olfactory recess. Pg: hypophysis. Po: pons. Sp: soft palate. Ta: thalamus. Th: telencephalon hemispherium (parietal lobe). ThC: cartilago thyroidea. Tm: tectum mesencephali.

**Figure 11 vetsci-12-00433-f011:**
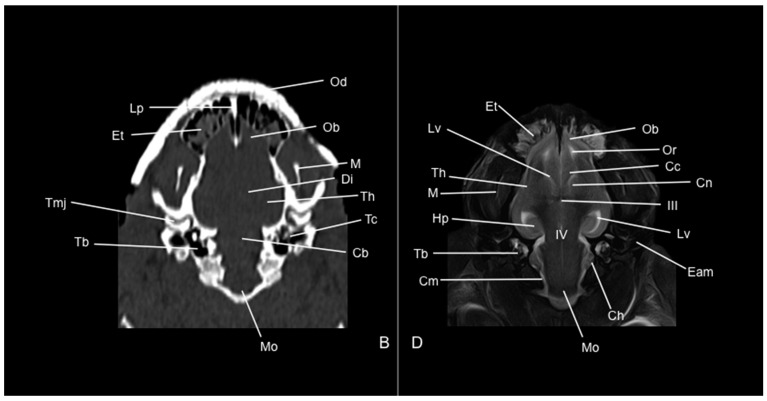
Dorsal CT bone window (**B**) and T2W (**D**) MR images of the head of a six-banded armadillo (armadillo 2) at the level of the auris media (middle ear). Cb: cerebellum (body). Cc: corpus callosum. Ch: cerebellum hemispherium. Cn: nucleus caudatus. Cm: cerebral meninges. Di: diencephalon. Eam: meatus acustici externi. Hp: hippocampus. III: third ventricle. IV: fourth ventricle. Lp: lamina perpendicularis (os ethmoidale). Lv: lateral ventricle. M: mandible. Mo: medulla oblongata. Ob: olfactory bulb. Od: osteoderm. Or: olfactory recess. Tb: bulla tympanica. Tc: cavum tympani. Th: telencephalon hemispherium. Tmj: articulatio temporomandibularis.

## Data Availability

The information is available at https://accedacris.ulpgc.es.

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
