# Peer review of "Comparative Analysis of Sectional Anatomy, Computed Tomography and Magnetic Resonance of the Cadaveric Six-Banded Armadillo (Euphractus sexcintus) Head"

_vetsci, 2025, doi:10.3390/vetsci12050433_

Round 1

Reviewer 1 Report

Comments and Suggestions for Authors

I think this manuscript is a very interesting topic. One disadvantage is that the sample size is small, there are only three samples. I think the author explained this limitation well in the manuscript

In this manuscript, they used three samples, was there the difference between the three samples. Such as a difference between male and female.

Author Response

Dear Reviewer,

We greatly appreciate your comments on our manuscript. As you correctly pointed out, the sample size is limited,  which prevents us from assessing potential differences between male and female specimens.

To our knowledge, previous anatomical studies in armadillos have not reported significant sex-based differences in the nervous system; however, we acknowledge that further research with larger sample sizes is needed to explore this possibility.

Reviewer 2 Report

Comments and Suggestions for Authors

The manuscript aimed to identify and describe the cranial structures of the six-banded armadillo using conventional cross-sectional anatomy, CT and MRI. It is well-written, with adequate background and strong rationale provided for the study.

Please find the following comments for your reference
Line 67 The six-banded armadillo is considered as a Least Concern species by the IUCN.
Line 118 ...1 mm slice thickness using a standard clinical protocol - should change the term "clinical" as it was considered as a postmortem CT instead.
Line 113/128 what is the condition of the carcass by the time the PMMRI was conducted? Was that done immediately after PMCT? Same question to PMCT as well, it will be appreciated to have the carcass condition and the preservation method listed as these would be affecting the image quality.
Line 396 exotic animal and wildlife medicine...
In the discussion, it would be nice to have a brief paragraph to compare and contrast the neuroimaging techniques and results of PMCT v PMMRI, reference can be made e.g. https://www.frontiersin.org/journals/marine-science/articles/10.3389/fmars.2020.544037/full, for the anatomical structure identifications.

Author Response

Dear Reviewer,

We sincerely appreciate your valuable comments and suggestions, which have greatly contributed to improving the quality of our manuscript.

- Comment 1, Line 67, The six-banded armadillo is considered as a Least Concern species by the IUCN.

We have modified the sentence.

- Comment 2, Line 118 ...1 mm slice thickness using a standard clinical protocol - should change the term "clinical" as it was considered as a postmortem CT instead.

As you recommended, we have deleted this term.

- Comment 3, Line 113/128 what is the condition of the carcass by the time the PMMRI was conducted? Was that done immediately after PMCT? Same question to PMCT as well, it will be appreciated to have the carcass condition and the preservation method listed as these would be affecting the image quality. 

Thank you for your thoughtful question. In this study, all three six-banded armadillo specimens were adult individuals that had died from causes unrelated to the central nervous system. The heads showed no clinical abnormalities, and the animals were in good postmortem condition at the time of the imaging studies. Although the exact postmortem interval is not specified in hours, the carcasses were preserved under appropriate conditions to ensure the quality of the imaging procedures.

CT and MRI examinations were performed sequentially in the same session, with the animals positioned in ventral recumbency for both modalities. This protocol ensured minimal time lapse between scans and consistent anatomical alignment. Immediately after imaging, the specimens were frozen at –80 °C for 72 hours to proceed with anatomical sectioning.

We recognize the influence of postmortem interval and temperature on image quality, particularly in MRI. However, the absence of notable autolysis artifacts and the excellent visualization of central nervous system structures in both T1- and T2-weighted images suggest that the preservation and imaging conditions were adequate. 

We appreciate your suggestion and will consider including further details on carcass handling and timing in future studies to enhance methodological transparency.

- Comment 4, Line 396 exotic animal and wildlife medicine.

The sentence has been modified following your recommendations.

- Comment 5, In the discussion, it would be nice to have a brief paragraph to compare and contrast the neuroimaging techniques and results of PMCT v PMMRI

Thank you for your comment. We have included a brief comparison between postmortem computed tomography (PMCT) and postmortem magnetic resonance imaging (PMMRI) in the discussion section, as both modalities provided complementary information in our study.

PMCT was particularly valuable for assessing skeletal structures and the presence of gas accumulations. It allowed a rapid and detailed evaluation of bone integrity, sinus cavities, and other air-containing spaces, which are essential in forensic and pathological investigations. PMCT also served as a useful reference to guide subsequent MRI acquisition and anatomical correlation.

PMMRI, on the other hand, offered superior contrast resolution for soft tissues, especially the central nervous system. It enabled the visualization of brain structures with sufficient detail to differentiate gray and white matter, ventricles, and even subtle anatomical landmarks. Despite being more sensitive to postmortem changes and temperature conditions, PMMRI proved highly effective in characterizing internal structures that were not visible or poorly defined in PMCT.

Therefore, both imaging modalities complemented each other: PMCT contributed structural and contextual information, while PMMRI enhanced soft tissue assessment, particularly of neural components.

Reviewer 3 Report

Comments and Suggestions for Authors

Thank you very much for this important work on the anatomy of the Euphractus sexcintus. I share the belief in the importance of anatomical studies of wild animals, particularly those associated with imaging exams. I have a few questions that could further improve the work:

1. The paper is well written, but I missed some applications of the information presented in the introduction and discussion sections. Why is it important to understand the anatomy of the head of this species? What are the main types of medical treatments for this species that correlate with the imaging exams used?

2. Was the manuscript approved by an Ethics Committee? Please provide the approval number.

3. Regarding the terminology used, why are some terms in Latin and others in English? What criteria were applied in choosing the terminology?

4. How long after death were the animals studied? Please describe this in the materials and methods section.

Kind regards,

Author Response

Dear Reviewer,

We sincerely appreciate your thoughtful comments on our manuscript. Your suggestions have been extremely helpful in enhancing the quality and clarity of our work.

Comment 1. The paper is well written, but I missed some applications of the information presented in the introduction and discussion sections. Why is it important to understand the anatomy of the head of this species? What are the main types of medical treatments for this species that correlate with the imaging exams used?

​Thank you for your insightful question. Understanding the head anatomy of the six-banded armadillo (Euphractus sexcinctus) is crucial for several reasons, both in clinical veterinary practice and in comparative anatomical research.

Therefore, we have included two paragraphs, one in the introduction and other in the discusion section, explaining the different applications of these modalities.

Comment 2. Was the manuscript approved by an Ethics Committee? Please provide the approval number.

In accordance with Spanish legislation, as well as European Union regulations, studies conducted on post-mortem animal tissues do not require prior approval from an Ethics Committee. This is established under Spanish Royal Decree 53/2013, of February 1, which regulates the Protection of Animals Used for Scientific or Educational Purposes. It is aligned with Directive 2010/63/EU of the European Parliament and of the Council of 22 September 2010 on the protection of animals used for scientific purposes.

Comment 3: Regarding the terminology used, why are some terms in Latin and others in English? What criteria were applied in choosing the terminology?

We appreciate this insightful comment. In our manuscript, we followed the guidelines of the Nomina Anatomica Veterinaria (NAV) for anatomical nomenclature. Latin terms were used for specific anatomical structures where standardized Latin terminology is well established and widely recognized in comparative anatomy, particularly in veterinary and anatomical atlases. English terms were used when referring to broader concepts, functional aspects, or when Latin equivalents are not commonly used in the literature or clinical context.

This mixed approach aims to balance anatomical precision with readability, and is consistent with the practice adopted in several recent anatomical and veterinary studies. To clarify this for readers, we have now included a brief explanation of our terminology criteria in the Materials and Methods section.

Comment 4. How long after death were the animals studied? Please describe this in the materials and methods section.

Thank you for your observation. As requested, we have added the relevant information to the Materials and Methods section. Specifically, the animals were rapidly frozen after death to preserve tissue integrity. One week later, they were thawed for imaging procedures and subsequently refrozen prior to anatomical sectioning. This approach is consistent with preservation protocols used in similar anatomical and imaging studies, ensuring adequate structural integrity for both MRI/CT analysis and anatomical correlation.

Reviewer 4 Report

Comments and Suggestions for Authors

I have carefully read the work presented by Raduan Jaber et al. The study is interesting due to the methodology it employs, but even more so because it is conducted on exotic animals, particularly on a species of Xenarthra whose biological characteristics are outstanding compared to other mammals.

The different sections of the paper are clear and informative. It is also valuable that the authors acknowledge their limitations, such as refraining from drawing certain conclusions due to the low number of animals. Nevertheless, I suggest that the authors clarify the basis for their anatomical descriptions (was it NAV?).

  • On the other hand, the figures are clear, though the caption of Fig. 1 should specify what I, II, etc., represent. Are they cutting planes or something else?
  • How was the correlation between radiological images and anatomical cross-sections validated? Was any superimposition software or quantitative measurement approach employed?

  • What specific criteria were used to define structures as "salient features" (e.g., were comparative analyses made with other xenarthrans or mammals)?

  • Regarding the "well-developed" tympanic cavity: Is this characteristic unique to Euphractus sexcinctus or common among armadillos? Were any quantitative measurements performed to compare its size relative to other cranial structures?

  • Concerning the brain morphology: Is the development of olfactory bulbs proportionally larger than in other mammals with similar ecological habits?

  • The study would benefit from comparative data with other species, including non-xenarthran mammals, to better contextualize the anatomical findings. Was this considered during the analysis?

The discussion not only addresses the study's limitations but also compares the findings with other xenarthran species. I believe the morphological data are relevant for mammals as a whole, significant for exotic animal veterinary medicine, and could also serve as a basis for further research—for example, on the nervous system of this highly peculiar group.

Comments on the Quality of English Language

Is good

Author Response

Dear Reviewer 4,

We really appreciate your comments on our study. As you pointed out, our sample size was quite limited,  which prevents us from drawing certain conclusions.

- Comment 1, the authors clarify the basis for their anatomical descriptions (was it NAV?). 

We appreciate this insightful comment. In our manuscript, we followed the guidelines of the Nomina Anatomica Veterinaria (NAV) for anatomical nomenclature. English terms were used when referring to broader concepts, functional aspects, or when Latin equivalents are not commonly used in the literature or clinical context.

This mixed approach aims to balance anatomical precision with readability, and is consistent with the practice adopted in several recent anatomical and veterinary studies. To clarify this for readers, we have now included a brief explanation of our terminology criteria in the Materials and Methods section.

- Comment 2, the caption of Fig. 1 should specify what I, II, etc., represent. Are they cutting planes or something else?

As you  recommended, we have included information about this. Therefore, we have added " Each number represents the approximate location for each CT and MR transverse (I to VIII) images.

- Comment 3, How was the correlation between radiological images and anatomical cross-sections validated? Was any superimposition software or quantitative measurement approach employed?

Thank you for your insightful question. CT images were used as a reference to guide the acquisition of MRI scans and to assist in the anatomical correlation process. Furthermore, anatomical cross-sections were carefully reviewed and compared with the corresponding radiological images by two experienced anatomists. Although no specific superimposition software or quantitative image registration tools were employed, the correlation was achieved through meticulous visual alignment based on anatomical landmarks across modalities. This approach ensured accurate matching between the imaging and anatomical planes.

- Comment 4, What specific criteria were used to define structures as "salient features" (e.g., were comparative analyses made with other xenarthrans or mammals)?

Thank you for your question. In our study, structures were considered “salient features” based on a combination of morphological prominence, clarity in imaging and anatomical sections, and their known functional or diagnostic relevance. While no formal morphometric or statistical comparative analysis was conducted, the identification and description of these features were guided by comparisons with existing anatomical literature on other xenarthrans and mammals, particularly the nine-banded armadillo and the giant anteater.

- Comment 5, Regarding the "well-developed" tympanic cavity: Is this characteristic unique to Euphractus sexcinctus or common among armadillos? Were any quantitative measurements performed to compare its size relative to other cranial structures?

Thank you for your thoughtful question. The description of the tympanic cavity as "well-developed" in Euphractus sexcinctus is based on its clear anatomical delineation and relative prominence in both CT and anatomical sections. While this characteristic is notable in our specimens, it is not unique to this species. A well-developed tympanic bulla has also been reported in other armadillo species, such as Dasypus novemcinctus and Chaetophractus villosus, as well as in related xenarthrans like the giant anteater, and even in non-xenarthran species such as guinea pigs and crested porcupines, where enhanced auditory structures are considered adaptations to their ecological niches.

However, no quantitative measurements were performed in this study to assess the size of the tympanic cavity relative to other cranial structures or to compare it across species. Our assessment was qualitative and descriptive, supported by previous anatomical references and comparative literature. We agree that future studies incorporating morphometric analysis across multiple xenarthran species would be valuable to further explore interspecific variations in tympanic cavity development.

- Comment 6, Concerning the brain morphology: Is the development of olfactory bulbs proportionally larger than in other mammals with similar ecological habits?

In our study, the olfactory bulbs of Euphractus sexcinctus appeared particularly prominent in both MRI and anatomical cross-sections, a finding that aligns with previous descriptions of xenarthrans, including Dasypus hybridus and Dasypus novemcinctus, where a similarly well-developed rhinencephalon has been reported. Although we did not conduct quantitative comparisons with other mammals of similar ecological habits (e.g., fossorial or nocturnal species), the observed prominence of the olfactory bulbs in our specimens supports the notion that olfaction plays a central role in the sensory ecology of this species.

This anatomical trait is consistent with the armadillo’s reliance on olfactory cues for foraging and predator detection, as reported in ecological studies. Therefore, while our assessment was descriptive, the relative size and organization of the olfactory bulbs appear proportionally larger compared to many generalist mammals, and are in line with what is expected in species that depend heavily on their sense of smell. We agree that further morphometric studies would be valuable to quantify these observations across species with similar ecological niches.

- Comment 7, The study would benefit from comparative data with other species, including non-xenarthran mammals, to better contextualize the anatomical findings. Was this considered during the analysis?

We acknowledge that incorporating quantitative comparative data with a broader range of mammals, including non-xenarthrans, would enhance the contextual understanding of our findings. Future studies employing morphometric analyses and advanced imaging techniques across diverse taxa could provide deeper insights into the evolutionary and functional implications of the anatomical features observed in Euphractus sexcinctus.